# Faster Neural Networks Straight from JPEG

**Lionel Gueguen**[1]    **Alex Sergeev**[1]    **Ben Kadlec**[1]    **Rosanne Liu**[2]    **Jason Yosinski**[2]

[1]Uber    [2]Uber AI Labs    {lgueguen,asergeev,bkadlec,rosanne,yosinski}@uber.com

## Abstract

The simple, elegant approach of training convolutional neural networks (CNNs) directly from RGB pixels has enjoyed overwhelming empirical success. But could more performance be squeezed out of networks by using different input representations? In this paper we propose and explore a simple idea: train CNNs directly on the blockwise discrete cosine transform (DCT) coefficients computed and available in the middle of the JPEG codec. Intuitively, when processing JPEG images using CNNs, it seems unnecessary to decompress a blockwise frequency representation to an expanded pixel representation, shuffle it from CPU to GPU, and then process it with a CNN that will learn something similar to a transform back to frequency representation in its first layers. Why not skip both steps and feed the frequency domain into the network directly? In this paper, we modify `libjpeg` to produce DCT coefficients directly, modify a ResNet-50 network to accommodate the differently sized and strided input, and evaluate performance on ImageNet. We find networks that are both faster and more accurate, as well as networks with about the same accuracy but 1.77x faster than ResNet-50.

## 1   Introduction

The amazing progress toward training neural networks, particularly convolutional neural networks [14], to attain good performance on a variety of tasks [13, 19, 20, 10] has led to the widespread adoption of such models in both academia and industry. When CNNs are trained using image data as input, data is most often provided as an array of red-green-blue (RGB) pixels. Convolutional layers proceed to compute features starting from pixels, with early layers often learning Gabor filters and later layers learning higher level, more abstract features [13, 27].

In this paper, we propose and explore a simple idea for accelerating neural network training and inference in the common scenario where networks are applied to images encoded in the JPEG format. In such scenarios, images would typically be decoded from a compressed format to an array of RGB pixels and then fed into a neural network. Here we propose and explore a more direct approach. First, we modify the `libjpeg` library to decode JPEG images only partially, resulting in an image representation consisting of a triple of tensors containing discrete cosine transform (DCT) coefficients in the YCbCr color space. Due to how the JPEG codec works, these tensors are at different spatial resolutions. We then design and train a network to operate directly from this representation; as one might suspect, this turns out to work reasonably well.

**Related Work**   When training and/or inference speed is critical, much work has focused on accelerating network computation by reducing the number of parameters or by using operations more computationally efficient on a graphics processing unit (GPU) [12, 3, 9]. Several works have employed spatial frequency decomposition and other compressed representations for image processing without using deep learning [22, 18, 8, 5, 7]. Other works have combined deep learning with compressed representations other than JPEG to promising effect [24, 1]. The most similar works to ours come from [6] and [25]. [6] train on DCT coefficients compressed not via the JPEG encoder but by a

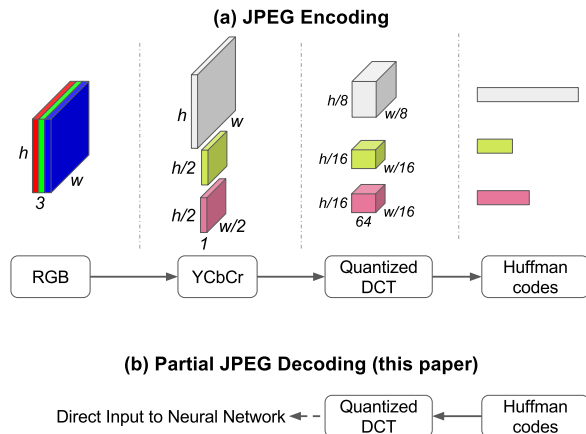

Figure 1: **(a)** The three steps to encode JPEG images: first, the RGB image is converted to the YCbCr color space and the *chroma* channels are downsampled, then the channels are projected through the DCT and quantized, and finally the quantized coefficients are losslessly compressed. See Sec. 2 for full details. **(b)** JPEG decoding follows the inverse process. In this paper, we run only the first step of decoding and then feed the DCT coefficients directly into a neural network. This saves time in three ways: the last steps of normal JPEG decoding are skipped, the data transferred from CPU to GPU is smaller by a factor of two, and the image is already in the frequency domain. To the extent early layers in neural networks already learn a transform to the frequency domain, this allows the use of neural networks with fewer layers.

simpler truncation approach. [25] train on a similar input representation but do not employ the full early JPEG stack, in particular not including the Cb/Cr downsampling step. Thus our work stands on the shoulders of many previous studies, extending them to the full early JPEG stack, to much deeper networks, and to training on a much larger dataset and more difficult task. We carefully time the relevant operations and perform ablation studies necessary to understand from where performance improvements arise.

The rest of the paper makes the following contributions. We review the JPEG codec in more detail, giving intuition for steps in the process that have features appropriate for neural network training (Sec. 2). Because the Y and Cb/Cr DCT blocks have different resolution, we consider different architectures inspired by ResNet-50 [10] by which the information from these different channels may be combined, each with different speed and performance considerations (Sec. 3 and Sec. 5). It turns out that some combinations produce much faster networks at the same performance as baseline RGB models or better performance at a more modest speed gain (Fig. 5). Having found faster and more accurate networks in DCT space, we ask whether one could simply find a nearby ResNet architecture that operates in RGB space that exhibits the same boosts to performance or speed. We find that simple mutations to ResNet-50 do not produce competitive networks (Sec. 4). Finally, given the superior performance of the DCT representation, we do an ablation study to examine whether this is due to the different color space or specific first layer filters. We find that the exact DCT transform works curiously well, even better than trying to learn a transform of the same dimension (Sec. 4.3, Sec. 5.3)! So others may reproduce experiments and benefit from speed increases found in this paper, we release our code at `https://github.com/uber-research/jpeg2dct`.

## 2 JPEG Compression

### 2.1 The JPEG Encoder

The JPEG standard (ISO/IEC 10918) was created in 1992 as the result of an effort started as early as 1986 [11]. Despite it being over 30 years old, the JPEG standard, which supports both 8-bit grayscale images and 24-bit color images, remains the dominant image representation in consumer electronics and on the internet. In this paper, we consider only the 24-bit color version, which begins with RGB pixels encoded with 8 bits per color channel.

As illustrated in Fig. 1a, JPEG encoding consists of the following three steps. The color space of an image is converted from RGB to YCbCr, consisting of one *luma* component (Y), representing the brightness, and two *chroma* components, Cb and Cr, representing the color. The spatial resolution of the *chroma* channels is reduced, usually by a factor of 2 or 3, while the resolution of Y is kept the same. This basic compression takes advantage of the fact that the eye is less sensitive to fine color details than to fine brightness details. In this paper, we assume a reduction by a factor of 2. Each of the three Y, Cb, and Cr channels in the image is split into blocks of 8×8 pixels, and each block undergoes a DCT, which is similar to a Fourier transform in that it produces a spatial frequency spectrum. The amplitudes of the frequency components are then quantized. Since human vision is much more sensitive to small variations in color or brightness over large areas than to the strength of high-frequency brightness variations, the magnitudes of the high-frequency components are stored with a lower accuracy than the low-frequency components. The quality setting of the encoder (for example 50 or 95 on a scale of 0–100 in the Independent JPE Group's library) affects the extent to which the resolution of each frequency component is reduced. If a very low-quality setting is used, many high-frequency components may be discarded as they end up quantized to zero. The size of the resulting data for all 8×8 blocks is further reduced using a lossless compression algorithm, a variant of Huffman encoding. Decoding or decompression from JPEG entails the corresponding inverse transforms in reverse order of the above steps; inverse transforms are lossless except for the inverse of the quantization step. Due to the loss of precision during the quantization of the DCT coefficients, the original image is recovered up to some distortions.

A standard implementation of the codec is `libjpeg` [15] released for the first time on 7-Oct-1991. The current version is the release 9b of 17-Jan-2016, and it provides a stable and solid foundation of the JPEG support for many applications. An accelerated branch, `libjpeg-turbo` [16], has been developed for exploiting Single Instruction Multiple Data (SIMD) parallelism. Other even faster versions have been developed that leverage the high parallelism of GPUs [23], where the Huffman codec is run on the CPU, and the pixel transformations, such as the color space transform and DCT, are executed on the GPU. Fig. 1 shows the JPEG encoding process and a schematic view of the partial decoding process we employ in this paper. We decode a compressed image up to its DCT coefficients, which are then directly inputted to a CNN. Because CNNs often compute Gabor filters on the first layer [13, 29, 28], and Gabor filters are similar to the conversion to frequency space realized by the DCT, it may be possible to prune the CNN of its first few layers without detriment; we experimentally verify this hunch in later sections. When using DCT coefficients, one has the option to either cast quantized values from int directly to float or to put them through the approximate inverse quantization process employed by the JPEG decoder. We chose to approximately invert quantization as it results in a network less sensitive to the quantization tables, which depend on the compression quality.

## 2.2   Details of the DCT Transform

Before delving into network details, it is worth considering a few aspects of the DCT in more detail. In JPEG compression, the DCT transform [17] is applied to non-overlapping blocks of size 8×8. Each block is projected onto a basis of 64 patterns representing various horizontal, vertical, and composite frequencies. The basis is orthogonal, so any block can be fully recovered from the knowledge of its coefficients. The DCT can be thought of as convolution with a specific filter size of 8×8, stride of 8×8, one input channel, 64 output channels, and specific, non-learned orthonormal filters. The 64 filters are illustrated in Fig. 2a. Let us consider a few details. Because the DCT processes each of the three input channels (one for *luminance* and two for *chroma*) separately, in terms of convolution it should be thought of as a three separate applications of convolution to three single-channel input images (equivalently: *depthwise* convolution), because information from separate input channels stays separate. Because the filter size and stride are both 8, spatial information does not cross to adjacent blocks. Finally, note that while the standard convolutional layer may learn an orthonormal basis, in general it will not. Instead, learned bases may be undercomplete, complete but not orthogonal, or overcomplete, depending on the number of filters and spatial size.

## 3   Designing CNN models for DCT input

In this section, we describe transforms that facilitate the adoption of DCT coefficients by a conventional CNN architecture such as ResNet-50 [10]. Some careful design is required, as DCT coefficients from the Y channel, $D_Y$, generally have a larger size than those from the *chroma* channels, $D_{Cb}$ and $D_{Cr}$, as shown in Fig. 1a, where the actual shapes are calculated based on an image input size of

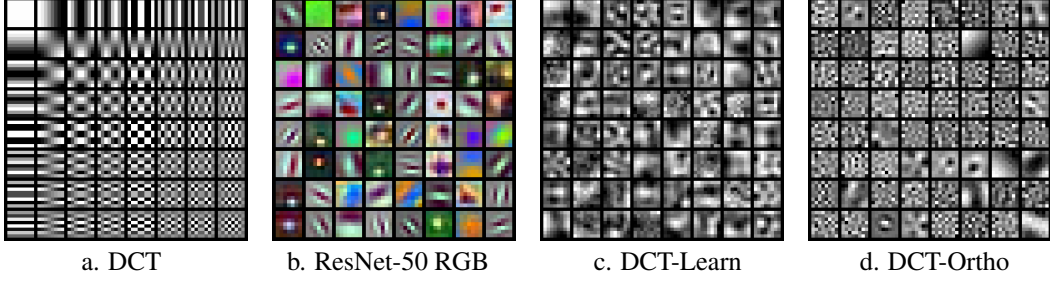

| a. DCT | b. ResNet-50 RGB | c. DCT-Learn | d. DCT-Ortho |

Figure 2: **(a)** The 64 orthonormal DCT basis vectors used for decomposing single-channel 8×8 pixel blocks in the JPEG standard [26]. **(b)** The 64 first-layer convolution filters of size 7×7 learned by a baseline ResNet-50 network operating on RGB pixels [10]. **(c)** The 64 convolution filters of size 8×8 learned starting from random weights by the DCT-Learn network described in Sec. 4.3. **(d)** The 64 convolution filters from the DCT-Ortho network, similar to (c) but with an added orthonormal regularization.

$224 \times 224$. It is necessary, then, to have special transforms that take care of the spatial dimension matching, before the resulting activations can be concatenated and fed into a conventional CNN. We consider two abstract transforms $(T_1, T_2)$ that separately operate on different coefficient channel, with the objective of resulting in matching spatial sizes among three activations $a_Y$, $a_{Cb}$ and $a_{Cr}$, where $a_Y = T_1(D_Y)$, $a_{Cb} = T_2(D_{Cb})$, and $a_{Cr} = T_2(D_{Cr})$. Fig. 3 illustrates this process.

In addition to ensuring that convolutional feature map sizes align, it is important to consider the resulting receptive field size and stride (hereafter denoted with $\mathcal{R}$ and $\mathcal{S}$) for each unit at the end of transforms and throughout the network. Whereas for typical networks taking RGB input, the receptive field and stride of each unit will be the same in terms of each input channel (red, green, blue), here the receptive fields considered *in the original pixel space* may be different for information flowing through the Y channel vs the Cb and Cr channels, which is probably not desired. We examine the representation size resulting from the DCT operation, and when compared with the same set of parameters of a ResNet-50 at various blocks (bottom table), we find that the spatial dimensions of $D_Y$ matches the activation dimensions of Block 3, while the spatial dimensions of $D_{Cr}$ and $D_{Cb}$ matches those from Block 4. This inspired us to skip some of the ResNet blocks in the design of network architecture, but skipping without further modification results in a much less powerful network (fewer layers and fewer parameters), as well as final network layers with much smaller receptive fields.

The transforms $(T_1, T_2)$ are generic and allow us to bring the DCT coefficients to a compatible size. In determining transforms we considered the following design concepts. The transforms can be (1) non-parametric and/or manually designed, such as up- or down-sampling of the original DCT coefficients, (2) learned, and can be simply expressed as convolution layers, or (3) a combination of layers, such as a ResNet block itself. We explored seven different methods of transforms $(T_1, T_2)$, from the simplest upsampling to deconvolution, and combined with different options of subsequent ResNet block stacks. We describe each, with further details in Sec. S1 in the Supplementary Information:

- UpSampling. Both *chroma* DCT coefficients $D_{Cb}$ and $D_{Cr}$ are upsampled by duplicating pixels by a factor of two in height and width to the dimensions of $D_Y$. The three are then concatenated channelwise, and go through a batch normalization layer before going into ResNet ConvBlock 3 (CB$_3$) but with reduced stride 1, then standard CB$_4$ and CB$_5$.

- UpSampling-RFA. This setup is similar to UpSampling, but here we keep ResNet CB$_2$ (rather than removing it) and CB$_2$ and CB$_3$ such that they mimic the increase in $\mathcal{R}$ and $\mathcal{S}$ observed in the original ResNet-50 blocks; we denote this "Receptive Field Aware" or RFA. As illustrated in Fig. 4, without this modification, the jump in $\mathcal{R}$ from input to the first block is large and the $\mathcal{R}$ later in the network is never as large (green line) as in the baseline ResNet. By instead keeping CB$_2$ but decreasing its stride, the transition to large $\mathcal{R}$ is more gradual and upon reaching CB$_3$ $\mathcal{R}$ and $\mathcal{S}$ match the baseline ResNet through the rest of the layers. The architecture is depicted in Fig. 3b and in Fig. S1.

- Deconvolution-RFA. An alternative to upsampling is a learnable deconvolution layer. In this design, we use two separate deconvolution layers on $D_{Cb}$ and $D_{Cr}$ to increase the spatial size. The rest of the design is the same as UpSampling-RFA.

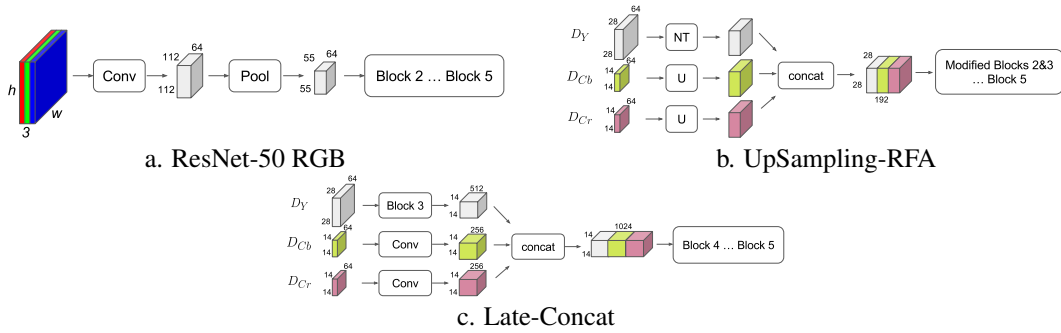

Figure 3: **(a)** The first layers of the original ResNet-50 architecture [10]. **(b)** The architecture of UpSampling-RFA is illustrated with coefficients $D_Y$, $D_{Cb}$ and $D_{Cr}$ of dimensions $28 \times 28 \times 64$ and $14 \times 14 \times 64$, respectively. The short names *NT* and *U* stand for the operations No Transform and Upsampling, respectively. **(c)** The architecture Late-Concat is depicted where the luminance coefficients $D_Y$ go through the ResNet Block 3, while the chroma coefficients go through single convolutions. This results in extra total computation along the luma path compared to the chroma path and tends to work well.

- DownSampling. As opposed to upsampling spatially smaller coefficients, another approach is to downsample the large one, $D_Y$, with a convolution layer. The rest of the design is similar to UpSampling, but with a few changes made to handle smaller input spatial size. As we will see in Sec. 5, this network operating on smaller total input results in much faster processing at the expense of higher error.

- Late-Concat. In this design, we run $D_Y$ on its own through two ConvBlocks (CBs) and three IdentityBlocks (IBs) of ResNet-50. In parallel, $D_{Cb}$ and $D_{Cr}$ are passed through a CB before being concatenated with the $D_Y$ path. The joined representation is then fed into the standard ResNet stack just after $CB_4$. The architecture is depicted in Fig. 3c and in Fig. S1. The effect is extra total computation along the luma path compared to the chroma path, and the result is a fast network with good performance.

- Late-Concat-RFA. This receptive field aware version of Late-Concat passes $D_Y$ through three CBs with kernel size and strides tweaked such that the increase in $\mathcal{R}$ mimics the $\mathcal{R}$ in the original ResNet-50. In parallel $D_{Cb}$ and $D_{Cr}$ take the same path as in Late-Concat before being concatenated to the result of the $D_Y$ path. The comparison of averaged receptive field is illustrated in Fig. 4, where one can see that Late-Concat-RFA has a smoother increase of receptive fields in comparison to Late-Concat. As explained in Fig. S1 for details, because the spatial size is smaller than in a standard ResNet, we use a larger number of channels in the early blocks.

- Late-Concat-RFA-Thinner. This architecture is identical to Late-Concat-RFA but with modified numbers of channels. The number of channels is decreased in the first two CBs along the $D_Y$ path and increased in the third, changing channel counts from {1024, 512, 512} to {384, 384, and 768}. The $D_{Cb}$ and $D_{Cr}$ components are fed through a CB with 256 channels instead of 512. All other parts of the network are identical to Late-Concat-RFA. These changes were made in an attempt to keep the performance of the Late-Concat-RFA model but obtain some of the speed benefits of the Late-Concat. As will be shown in Fig. 5, it strikes an attractive balance.

## 4 RGB Network Controls

As we will observe in Sec. 5 and Fig. 5, many of the networks taking DCT as input perform with lower error and/or higher speed than the baseline ResNet-50 RGB. In this section, we examine whether this is just due to making many architecture tweaks, some of which happen to work better than a baseline ResNet. Here we start with a baseline ResNet and attempt to mutate the architecture slightly to get it to perform with lower error and/or higher speed. Inputs are RGB images of size $224 \times 224 \times 3$.

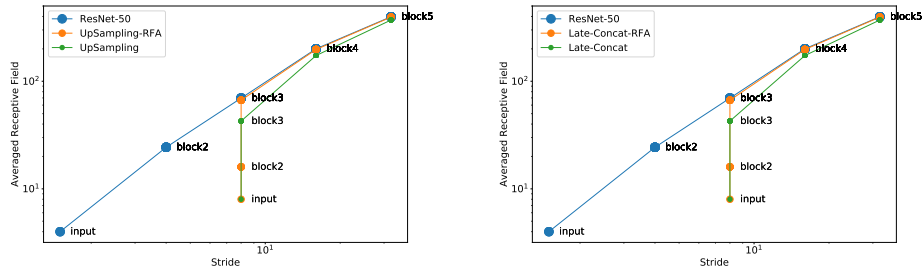

Figure 4: The average of receptive field sizes within each ResNet block vs. the corresponding block stride. Both axes are in log scale. The measurements are reported for some of the DCT based architectures, and they are compared to the growth of the receptive field observed in ResNet-50. The plots underline how the receptive field aware (RFA) versions of basic DCT based architectures allow a transition similar to the one observed in the baseline network.

## 4.1 Reducing the Number of Layers

To start, we test the simple idea of removing convolution layers in ResNet-50. We remove the Identity blocks one at a time, from the bottom up, from Blocks 2 and 3, resulting in 6 experiments as 6 layers are removed. We never remove the convolution layer between Blocks 2 and 3 to keep the number of channels in each block and representation size unchanged.

In this series of experiments, the first identity layer (ID) from Block 2 is removed first. Secondly, both the first and second ID layers are removed. The experiment continues until all 3 ID layers of both Block 2 and 3 are removed. In the last configuration, the network shares similarities with the UpSampling architecture, where the RGB signal is transformed with a small number of convolutions to a representation size of $28 \times 28 \times 512$. The RGB input goes through the following series of layers: convolution, max pooling, one last identity layer from Block 3. We can see the trade-off between the inference speed and accuracy in Fig. 5 under the legend "Baseline, Remove ID Blocks" (series of 6 gray squares). As shown, networks become slightly faster but at a large reduction in accuracy.

## 4.2 Reducing the Number of Channels

Because reducing the number of layers worked poorly, we also investigate thinning the network: reducing the number of channels in each layer to speed up inference. The last fully connected layer is modified to adapt to the size of its input layer while maintaining the same number of outputs. We propose to reduce the number of channels by taking the original number of channels and dividing it by a fixed ratio. We conduct three experiments with ratios $\{1.1, \sqrt{2}, 2\}$. The same trade-off between speed or GFLOPS and accuracy is shown in Fig. 5 under the legend "Reduced # of Channels". As with reducing the number of layers, networks become slightly faster but at a large reduction in accuracy. Perhaps both results might have been suspected, as the authors of ResNet-50 likely already tuned the network depth and width well; nevertheless, it is important to verify that the performance improvements observed could not have been obtained through this much simpler approach.

## 4.3 Learning the DCT Transform

A final set of four experiments — shown in Fig. 5 as four "YCbCr pixels, DCT layer" diamonds — address whether we can obtain a similar benefit to the DCT architectures but starting from RGB pixels by using convolutional layers designed to replicate, exactly or approximately, the DCT transform. RGB images are first converted into YCbCr space, then each channel is fed independently through a convolution layer. To mimic the DCT, the convolution filter size is set to 8×8 with a stride of 8, and 64 output channels (or in some cases: more) are used. The resulting activations are then concatenated before being fed into ResNet Block 2. In DCT-Learn, we randomly initialize filters and train them in the standard way. In DCT-Ortho, we regularize the convolution weights toward orthonormality, as described in [2], to encourage them not to discard information, inspired by the orthonormality of the DCT transform. In DCT-Frozen, we simply use the exact DCT coefficients without training, and in DCT-Frozenx2 we modify the stride to be 4 instead of 8 to increase representation size at that layer and allow filters to overlap. Surprisingly, this network tied the performance (6.98%) of the best other

Table 1: The averaged top-1 and top-5 error rates are represented for the baseline ResNet-50 architecture and the proposed DCT based ones. Standard deviation is appended to the top error rates for experiments repeated more than three times. The frame per second inference speed measured on an NVIDIA Pascal GPU is also reported given that data is packed in batches of size 1024.

| ARCHITECTURE | TOP-1 ERR | TOP-5 ERR | TOP-1 DIFF | TOP-5 DIFF | FPS |
|---|---|---|---|---|---|
| RESNET-50 RGB | $24.22 \pm 0.08$ | $7.35 \pm 0.004$ | - | - | 208 |
| RESNET-50 YCBCR | 24.36 | 7.36 | +0.14 | +0.01 | 207 |
| UPSAMPLING | $25.07 \pm 0.07$ | $7.81 \pm 0.12$ | +0.85 | +0.45 | 396 |
| UPSAMPLING-RFA | $24.06 \pm 0.09$ | $7.14 \pm 0.07$ | -0.16 | -0.21 | 266 |
| DECONVOLUTION-RFA | $\mathbf{23.94} \pm 0.015$ | $\mathbf{6.98} \pm 0.005$ | **-0.27** | **-0.36** | 268 |
| DOWNSAMPLING | 27.00 | 8.98 | +2.78 | +2.36 | **451** |
| LATE-CONCAT | 24.93 | 7.62 | +0.71 | +0.27 | 401 |
| LATE-CONCAT-RFA | 24.08 | 7.09 | -0.14 | -0.25 | 267 |
| LATE-CONCAT-RFA-THINNER | 24.61 | 7.43 | +0.39 | +0.08 | 369 |

approach when averaged over three runs, though without the speedup of the Deconvolution-RFA approach. This is interesting because it departs from network design rules of thumb currently in vogue: first layer filters are large instead of small, hard-coded instead of learned, run on YCbCr space instead of RGB, and process channels depthwise (separately) instead of together. Future work could evaluate to what extent we should adopt these atypical choices as standard practice.

## 5   Results and Discussions

Experiments described in Section 3 and 4 are conducted with the Keras framework and TensorFlow backend. Training is performed on the ImageNet dataset [4] with the standard ResNet-50 stepwise decreasing learning rates described in [10]. The distributed training framework Horovod [21] is employed to facilitate parallel training over 128 NVIDIA Pascal GPUs. To accommodate the parallel training, the learning rates are multiplied by the number of parallel running instances. Each experiment trains for 90 epochs, which correspond to only 2-3 hours in this parallelization setup. A total of more than 50 experiments are run. All experiments are conducted with images which are first resized to 224×224 pixels with a random crop, and the JPEG quality used during encoding is 100, so as little information is lost as possible. A limitation of using a JPEG representation during training is that to do data augmentation e.g. via random crops, one must decompress the image, transform it, and then re-encode it before accessing the DCT coefficients. Of course, inference after the model is trained will not require this process. Inference time measurements are calculated by running the inference on 20 batches of size $1024 \times 224 \times 224 \times 3$ on the 128 GPUs where the overall time is collected, and the effective number of images per second per GPU is then calculated. All timing is computed for inference, not training, and is computed as if data were already loaded; thus timing improvements do not include possible additional savings due to reduced JPEG decompression time.

### 5.1   Error Rate versus Inference Speed

We report full results in Table 1, for all seven proposed DCT architectures from Section 3, along with two baselines: ResNet-50 on RGB inputs, and ResNet-50 on YCbCr inputs. The full results include validation top-1 and top-5 error rates and inference frames per second (FPS). Both ResNet baselines achieve a top-5 error rate of 7.35% at an inference speed of 208 FPS on an NVIDIA Pascal GPU, while the best DCT network achieves it at 6.98% with 268 FPS. We analyze the 7 experiment results by dividing them into three categories. The first category contains those where DCT coefficients are directly connected with the ResNet-50 architecture; this includes UpSampling, DownSampling, and Late-Concat. Several of these architectures providing significant inference speed-up (three far-right dots in Fig. 5), almost $2\times$ in the best case.

The speedup is due to less computation as a consequence of reduced ResNet blocks. A sharp increase of error with DownSampling suggests that a reduction in the spatial structure of the Y (*luma*) causes a reduction of information while maintaining its spatial resolution (as in UpSampling and Late-Concat) performs closer to the baseline. In the second category, the two best architectures above are extended to increase their $\mathcal{R}$ slowly, so as to mimic the $\mathcal{R}$ growth of ResNet-50 (see Fig. 4). This category contains UpSampling-RFA and Late-Concat-RFA, and they are shown to achieve better error rates

than their non-RFA counterparts while still providing an average speed-up of $1.3\times$. With the proper RFA adjustments in architecture, these two versions manage to beat the RGB baseline. A third category attempts to further improve the RFA architectures, by (1) learning the upsampling operation with Deconvolution-RFA, and (2) reducing the number of channels with Late-Concat-RFA-Thinner.

On the one hand, Deconvolution-RFA reduces the top-5 error rate of UpSampling-RFA by $0.15\%$ while maintaining an equivalent inference speed. On the other hand, Late-Concat-RFA-Thinner achieves error rates on par with the baseline while providing a speed-up ratio of $1.77\times$. A review of the GFLOPS for each architecture (cf. Fig. 5) shows that despite more computation of some architectures, all architectures achieve higher speeds thanks to halved data transfer between CPU and GPU. Speed tests performed for the Late-Concat-RFA architecture that ignore data transfer gains show that about 25% of the measured gain is due to limited data transfer.

## 5.2 Ability to Trade-Off

In analyzing results from RGB network controls, we observe a continual increase in inference speed and GFLOPS coupled with an increase in error rates, as the network size is reduced. None of the controls can maintain one while improving the other. The curves (gray and light gray in Fig. 5), however, exhibit how the two opposing forces play with each other and provide insights to the user to determine the trade-offs when choosing network size. We observe that decreasing the number of channels offers the worst trade-off curve, as the error rate increases drastically for only small speed-up gains. Removing the identity blocks offers a better trade-off curve, but this approach still allows only limited speed-ups and reaches a cliff where speed-up is bounded.

Considering the trade-off curves from DCT architectures (blue and red curves in Fig. 5), however, we notice the apparent advantage especially if one urges to gain an improvement on inference speed. We notice the significant gain in speedup while maintaining an error rate within a 1% range of the baseline. We conclude therefore that making use of DCT coefficients in CNNs constitutes an appealing strategy to balance loss versus computational speed. We also want to highlight two of the proposed DCT architectures that demonstrate compelling error/speed trade-offs. First, the Deconvolution-RFA architecture achieves the smallest top-5 error rate overall, while still improving inference speed by $30\%$ over the baseline (black square in the figure). Secondly, the Late-Concat-RFA-Thinner architecture provides an error rate closest to the baseline while allowing $77\%$ faster inference. Moreover, the small slopes of the two curves strongly manifest that at a slight cost of computation, the RFA tweaks in the design improves accuracy by allowing a slow, smooth increase of receptive fields.

## 5.3 Learning DCT Transform

Another interesting curve to examine is the result from Sec. 4.3, experiments attempting to learn convolutions behaving like DCT. It is the darker gray curve in Fig. 5 annotated with legends starting with "YCbCr pixels". The first two experiments trying to learn the DCT weights from random initialization, with and without orthonormal regularization, achieve slightly higher error rates than our RGB and DCT baselines. The third and fourth experiments relying on frozen weights initialized from the DCT filters themselves achieve lower error rates, on par with the best DCT architecture. These results show that learning the DCT filters is hard with one convolution layer and produce sub-performant networks. Moreover leveraging directly the DCT weights allow better error rates, making the JPEG DCT coefficients an appealing representation for feeding CNN.

## 6 Conclusions

In this paper, we proposed and tested the simple idea of training CNNs directly on the blockwise discrete cosine transform (DCT) coefficients computed as part of the JPEG codec. Results are compelling: at a similar performance to a ResNet-50 baseline, we observed speedups of 1.77x, and at performance significantly better than the baseline, we obtained speedups of 1.3x. This simple change of input representation may be an effective method of accelerating processing in a wide range of speed-sensitive applications, from processing large data sets in data-centers to processing JPEG images locally on mobile devices.

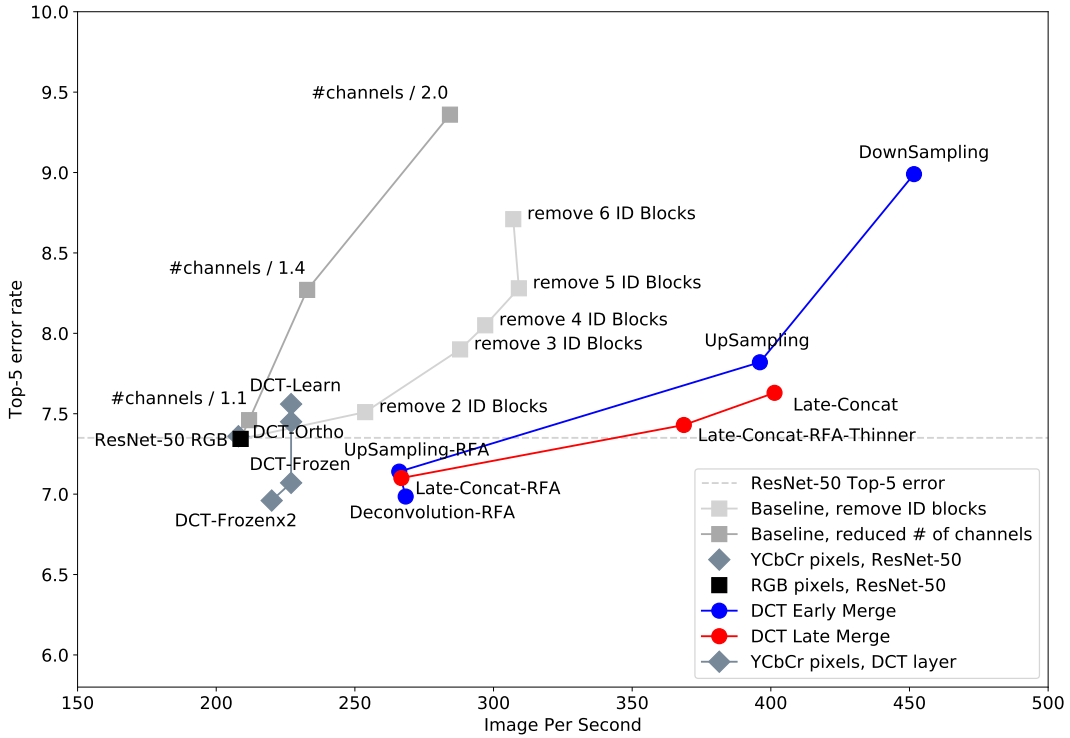

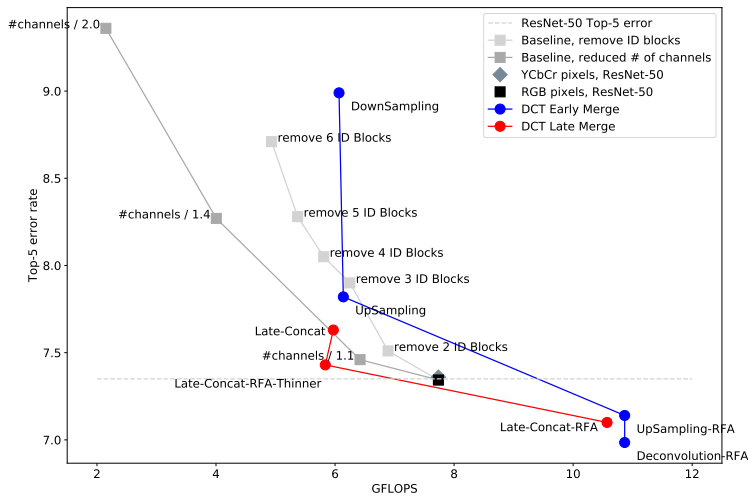

Figure 5: **(top)** Inference speed vs top-5 error rates. **(bottom)** GigaFLOPS vs top-5 error rates. Six sets of experiments are grouped. ResNet-50 baseline on both RGB and YCbCr show nearly identical performance, indicating that the YCbCr color space on its own is not sufficient for improved performance. Two sets of controls on the RGB baseline — baseline with removed ID blocks and with a reduced number of channels — show that simply making ResNet-50 shorter or thinner cannot produce speed gains at a competitive level of performance to the DCT networks. Finally, two sets of DCT experiments are shown, those that merge Y and Cb/Cr channels early in the network (within one layer of each other) or late (after more than a layer of processing of the Y channel). Several of these networks are both faster and more accurate, and the Late-Concat-RFA-Thinner network is about the same level of accuracy while being 1.77x faster than ResNet-50.

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

# Supplementary Information for:
# Faster Neural Networks Straight from JPEG

## S1 Details of model architectures

Fig. S1 shows the baseline ResNet-50 architecture as well as the seven architectures discussed in Sec. 3 that take DCT input.

| Baseline | UpSampling | UpSampling-RFA | Deconvolution-RFA | DownSampling | Late-Concat | Late-Concat-RFA |
|---|---|---|---|---|---|---|
| | Reference: Baseline | Reference: Upsampling | Reference: Upsampling-RFA | Reference: Baseline | Reference: Baseline | Reference: Baseline |
| RGB pix *(224, 224, 3)* | Y *(28, 28, 64)* Cb,Cr *(14, 14, 128)* | | Y *(28, 28, 64)* Cb,Cr *(14, 14, 128)* | Y *(28, 28, 64)* Cb,Cr *(14, 14, 128)* | Y *(28, 28, 64)* Cb,Cr *(14, 14, 128)* | Y *(28, 28, 64)* Cb,Cr *(14, 14, 128)* |
| C(64, 7, 2) | U *(28, 28, 128)* | | Deconv *(28, 28, 128)* | C(256, 2, 2) *(14, 14, 256)* | BN BN | BN BN |
| BN, R | | | | | $CB_4$(s=1) $CB_4$(k=1, s=1) | $CB_4$(k=1, s=1) $CB_4$(k=1, s=1) |
| M(3, 2) | Concat *(28, 28, 192)* · · · · | | Concat *(28, 28, 192)* · · · · · · | Concat *(14, 14, 192)* · · · · · · · · | IB, IB, IB | IB(k=2), IB |
| $CB_2$(s=1) | BN | → | ← | BN | $CB_4$ | $CB_3$(s=1) |
| IB, IB | | $CB_4$(k=1, s=1) | | $CB_4$(k=1, s=1) | Concat · · · · · · · · · · · · · · · · | IB, IB, IB |
| | | IB(k=2), IB | | IB(k=2), IB | | $CB_4$ |
| $CB_3$ | $CB_3$(s=1) | ← | | $CB_3$(s=1) | | Concat · · · · · · · · · · · · · · |
| IB, IB, IB | ← | | | — | | |
| $CB_4$ | | | | $CB_4$(s=1) | | |
| IB, IB, IB, IB, IB | | | | ← | ← | ← |
| $CB_5$ | | | | | | |
| IB, IB | | | | | | |
| GAP | | | | | | |
| FC(1000) | | | | | | |
| Softmax | | | | | | **Late-Concat-RFA-Thinner** |
| | | | | | | (Same as Late-Concat-RFA but with different number of channels; see text. |

| Legend | | | | |
|---|---|---|---|---|
| RGB pix | RGB pixel input | | $CB_n$ | ConvBlock stage n, with number of channels as in original ResNet-50 paper, kernel size = 3 and stride = 2 unless specified otherwise. |
| Y | Y-channel DCT input | | | |
| Cb, Cr | Cb- and Cr-channel DCT input | | IB | IdentityBlock, with number of channels matched to preceding CB layer (as in ResNet-50) |
| C | Convolution(channels, filter size, stride) | | | |
| Deconv | Deconvolution with 64 output channels, filter size 2, stride 2. Separate deconvolution layers are applied to Cb and to Cr, resulting in 128 total output channels. | | GAP | Global average pooling layer |
| | | | FC | Fully connected layer (channels) |
| BN | BatchNormalization | | Softmax | Softmax nonlinearity |
| R | Relu | | | |
| M | MaxPooling(pool size, stride) | | → | Layers up to this point are the same as reference |
| U | Upsampling layer (2x) | | ← | Layers after this point are the same as reference |
| Concat | Channelwise concatenation | | — | This layer or these blocks are same as reference |
| | Shape of representation at layer shown like this: *(height, width, channels)* For example: *(14, 14, 128)* | | | |

Figure S1: The baseline ResNet-50 architecture and the seven related architectures discussed in Sec. 3. Gray banded highlights are arbitrary and solely for visual clarity. The baseline ResNet-50 contains ConvBlocks $CB_1$, $CB_2$, $CB_3$, $CB_4$ with doubling number of channels at each stage increase. In this figure we use ConvBlock subscripts to refer to a block with the same number of channels as in ResNet-50, not to indicate the order of the CB within our model. Thus, for example, in the DownSampling model, $CB_4$ is followed by $CB_3$, another $CB_4$, and $CB_5$. Because models taking DCT input start with a representation with much lower spatial size but many more input channels, using ConvBlocks with many channels early in the network is advantageous. Best viewed electronically with zoom.

