[Reviews · NeurIPS 2018]

Reviewer 1



My main concerns were responded to in the the rebuttal (separation between IO/CPU/GPU gains), TFLOP measurements and discussion of related works. I was happy to see that their model (Late-Concat-RFA-Thinner) is still faster than ResNet 50 (approx 680/475= 43% gains in Fig. 1. c) rebuttal instead of 365/205= 78% in Fig. 4. paper) when feeding the models with zeros disregarding JPEG decoding (CPU time) and IO costs. This is a pessimistic estimate given that the ResNet 50 RGB needs to also do the inverse DCT to go from DCT coefficients to RGB domain. However, I was a bit surprised to see such a big disconnect between the timing numbers and the TFLOP measurements (Fig. 1. b vs Fig 1. c rebuttal). While I trust that the authors timed the models fairly and thus I do not doubt the results, I think this would be worth more investigation. For the related works, the authors did a good discussion of them in the rebuttal, but I find it strange that we had to ask for this. (This is something that should obviously be in the initial version of the paper!). Overall however, accounting for everything, this sentence in the rebuttal resonated with me: "We suggest that without our paper, anyone wanting to run modern deep networks on JPEG representations would first have to re-discover or re-engineer all these critical details that comprise the majority of our paper." Thus I'm keeping my rating to accept the paper. ================================ The paper presents a detailed study on a simple but old idea: given that most images are stored as JPEGs, and that the JPEG decoder is nothing but a linear transformation over 8x8 blocks, can we not avoid the JPEG decoding step? There are prior works that have explored similar ideas (which should be properly cited and discussed!), like: * On using CNN with DCT based Image Data Matej Ulicny & Rozenn Dahyot, IMVIP 2017 * Using Compression to Speed Up Image Classification in Artificial Neural Networks Dan Fu, Gabriel Guimaraes, 2016 * Towards Image Understanding from Deep Compression Without Decoding, Torfason et al. ICLR 2018 * High speed deep networks based on Discrete Cosine Transformation, Zou et al, ICIP 2014. However, aside from Torfason et al. (which do not focus on JPEG but deep compression), these papers are limited in the setting (not going far beyond MNIST classificatoin). In contrast, this paper thoroughly studies it in a large scale setting, showing practical gains for ImageNet classification, exploring the architecture space, e.t.c. However, I still have some concerns about the paper: * Related works needs more discussion * FPS measurements would be well complemented with TFLOP numbers Another concern I have is the following: are we seeing implementation speedups or computational speedups? The JPEG decoder is decoding on CPU, whereas the the network is on the GPU. By feeding DCT coefficients to the network, you reduce the CPU workload, so are we just seeing speedups from this workload being moved over to the GPU? The proper way to compare with ResNet-50 RGB would be to use the modified libjpeg created by the authors, and then implement the DCT - > RGB decoding step with a fixed 8x8 deconvolution (carefully checking that it gives the same output as standard JPEG decoder). This should speed up the ResNet50 RGB model, no? Overall the paper is well written and easy to follow and presents speedup on a standard classification pipeline that are significant.

Reviewer 2



The paper proposes the training of CNNs for classification directly on the DCT coefficients from the JPEG codec. The removal of the jpeg decoding and the work of CNN directly on DCT which allow for skipping first layers lead to speed advantages, reportedly 1.77x speed up of the modified net over the ResNet-50 baseline. The paper reads mostly well and the derived CNNs and experimental results support the idea of faster CNNs straight from JPEG DCT coefficients. The paper suffers, however, on a number of aspects. 1) Related work: I do not see a related work review of publications employing directly the compressed domain for higher level tasks. This work should be placed into the existing literature. 2) Novelty: This paper is not the first to propose working on the compressed domain of an image. Nor the first paper using CNN for classification on the compressed domain. 3) Generality: I am curious if the speed benefits obtained for the CNNs working for classification directly on the DCT coefficients are also achievable for other vision tasks such as (semantic) image segmentation, image translation.. I enumerate here some relevant literature for this paper: Shen et al, Direct feature extraction from compressed images, 1996 Mandal et al, A critical evaluation of image and video indexing techniques in the compressed domain, 1999 Hafed et al, Face Recognition Using the Discrete Cosine Transform, 2001 Feng et al, JPEG compressed image retrieval via statistical features, 2003 and more recent works: Torfasson et al, Towards Image Understanding from Deep Compression without Decoding, 2018 Adler et al, Compressed Learning: A Deep Neural Network Approach, 2016 Fu et al, Using compression to speed up image classification in artificial neural networks, 2016 Aghagolzadeh et al, On hyperspectral classification in the compressed domain, 2015

Reviewer 3



This paper proposes to train cnn directly from the second stage of JPEG decoding. - One crucial factor for getting good performance of CNN is to apply data augmentations(crop, rotate). It is unclear how the augmentations can be applied to the new pipeline - It would be helpful to characterize the gains of GPU inference time without considering IO, as the FPS number can depend on implementation. - The comprehensive study of various choices of applying pre-transform pipeline on the intermediate is useful and can be interesting to a lot of audiences. In summary, this paper presents interesting results that are relevant to NIPS audience. There are a few technicals points that can be improved to make the paper stronger. UPDATE ---------- I have read the rebuttal and will keep my score as it is. I encourage the author is encouraged to include the discussion on the impact of IO and the implication of data augmentation.